# Residents need competence not confidence: A retrospective evaluation of the new competency education program for Korean neurology residents

Hojin Choi[1], Jeeyoung Oh[2‡], Chi Kyung Kim[3‡], Hokyoung Ryu[4]*, Youngji Ryu[5]

1 Department of Neurology, Hanyang University, Seoul, Republic of Korea, 2 Department of Neurology, Konkuk University School of Medicine, Seoul, Republic of Korea, 3 Department of Neurology, Korea University Guro Hospital, Seoul, Republic of Korea, 4 Graduate School of Innovation & Technology Management, Hanyang University, Seoul, Republic of Korea, 5 School of Psychology, Korea University, Seoul, Republic of Korea

☯ These authors contributed equally to this work.
‡ JO and CKK also contributed equally to this work.
* hryu@hanyang.ac.kr

**Data Availability Statement:** Data cannot be shared publicly because of its nature being interview and survey data. A quantifiable data are available from the Hanyang University's

## Abstract

The objective of our study was to scrutinize the learning experiences of Korean neurology residents, with an emphasis on the implications of the novel competency-based curriculum implemented in 2021. We hypothesized that this revised curriculum could modulate residents' cognitive conduct, primarily the manifestation of overconfidence, in distinctive ways across different stages of training. Our investigative framework was three-fold. Initially, we began with a qualitative inquiry involving in-depth interviews with a purposively selected cohort of eight residents from four training sites. This approach facilitated comprehensive insight into their perceptions of their competence and confidence across the continuum of a four-year residency program. Subsequently, we incorporated the K-NEPA13 assessment instrument, administered to the residents and their overseeing supervisors. This stage aimed to dissect potential cognitive biases, particularly overconfidence and consistency, within the resident population. The final study involved a comprehensive survey administered to a group of 97 Korean neurology residents, allowing us to consolidate and validate our preceding findings. Our findings revealed that junior residents portrayed heightened confidence in their clinical capabilities compared to their senior peers. Intriguingly, junior residents also displayed a stronger inclination towards reevaluating their clinical judgments, a behavior we hypothesize is stimulated by the recently introduced competency-based curriculum. We identified cognitive divergence between junior and senior residents, with the latter group favoring more consistent and linear cause-and-effect reasoning, while the former demonstrated receptiveness to introspection and reconsideration. We speculate this adaptability might be engendered by the supervisor assignment protocol intrinsic to the new curriculum. Our study highlights the essentiality of incorporating cognitive behaviors when devising medical education strategies. Acknowledging and addressing these diverse

Institutional Data Access / Ethics Committee
(contact via rlaguseh12@hanyang.ac.kr, Data
Controller of Department of Intelligence
Computing, Hanyang University) for researchers
who meet the criteria for access to confidential
data.

**Funding:** CH/RH - Grant number: HC22C0014
Korea Health Industry Development Institute
(KHIDI) The Ministry of Health & Welfare, Republic
of Korea No. The funders had no role in study
design, data collection and analysis, the decision to
publish, or preparation of the manuscript.

**Competing interests:** The authors have declared
that no competing interests exist.

cognitive biases, and instilling a spirit of adaptability, can nurture a culture that persists in
continuous learning and self-reflection among trainee doctors.

## Introduction

Against the backdrop of the World Health Organization's urgent call for transformation in
healthcare education [1] due to anticipated workforce shortages, competency-based education
(CBE) has emerged as one of the prominent paradigms in the reconfiguration of resident education
systems. This shift moves from traditional time-based models towards a framework that
prioritizes demonstrated competency [2, 3]. Our study contributes significantly to this field by
providing an in-depth evaluation of the Competency-Based Medical Education (CBME) program
recently applied to Korean neurology residents.

Our research thus bridges the gap between theory and practice in resident education by
assessing how theoretical constructs, such as competency development and the associated
learning processes, translate into real-world settings. More important, we investigate the relationship
between self-assessed competence and actual performance, a critical aspect of CBE
that directly impacts patient care.

In this regard, a key theoretical construct to our study is 'pretense-competence,' where residents
might overestimate their own abilities due to fragmented or incomplete training [4–6].
Recent studies have thus highlighted the importance of supervisors intervening when trainee
doctors exhibit pretense-competence [6–10]. By being aware of the level of residents' confidence,
supervisors are better equipped to make informed guidance and give residents a certain
level of autonomy they need to perform in clinical contexts. Similar but a rather differing
nuance, Canady and Larzo [11] claimed that overconfidence made more problematic engagement
in clinician's health behaviors whilst not realizing this deficit [12–14]. By exploring this,
our study adds depth to the impact on clinical competency in the context of CBME, where we
can highlight the theoretical foundations of CBME and their practical insights to enhance
training quality and effectiveness.

While our focus is on the Korean neurology resident training program, the insights gained
from our study also have profound implications for medical education globally [15, 16]. Our
exploration of the CBME model applied in the Korean context offers lessons and provides evidence-based
recommendations for countries considering a similar shift in their resident education
programs. As such, our research contributes to the broader literature on effective
education strategies for healthcare professionals worldwide.

However, note that our study navigates the complex terrain of CBME without a direct comparison
to the previous curriculum, which is impossible to implement in the residents' continuing
education practically. Instead, we explore how the Korean CBME model has affected
resident learning experiences. This nuanced understanding contributes to a broader discourse
on the refinement of medical education strategies, aiming to optimize these outcomes.

## Methods

### Design

To empirically examine residents' competency (within the purview of CBME), one ideally
needs to test real clinical decisions and activities with a fully controlled or semi-experimental
approach. However, there are few articles in the medical education literature in which numerous
trainee doctors are randomly assigned for comparison purposes [17, 18]. Practically, it is

not possible for the patient's safety. Instead, this article intertwines three studies that compare the perceptions of competence between junior and senior neurology residents in Korea. The junior residents, comprising first and second-year residents, have undergone comprehensive training through the new CBME curriculum, whereas the senior residents have received training based on both the old and new curricula.

- Study 1: Eight trainee doctors (four junior and four senior) were individually interviewed for around two hours and asked to reflect on their clinical responsibilities and decision-making, including some failure cases.

- Study 2: A total of eight supervisors and 44 residents from five university hospitals (HU, KKU, KU, IU, and CNNU in Korea) evaluated the residents' competence level on K-NEPA13 (Korean Neurologist's Entrustable Professional Activities 13, see S1 Appendix). The study was designed to compare the residents' self-evaluation of their clinical competence to the supervisors' evaluation.

- Study 3: A survey of 97 residents was conducted to see how the new competence curriculum in the resident training changed the trainee doctor's mindset (such as preparedness, competencies for independence, and thinking style, see S2 Appendix).

The studies were approved and secured by the human research ethics committee of Hanyang University, South Korea (Research ethics GURI 2022-08-023-001) and adhered to the tenets of the Declaration of Helsinki.

## Sampling strategy, setting, and data collection

Given the potentially sensitive nature of the research questions, the first study conducted in-depth interviews individually, for around two hours. To ease in and build rapport, the interview protocol began with questions about learning from recent patient care, where they found it challenging to tell the appropriate clinical decisions. After asking about their personally difficult experiences of the recent patient cases, we asked about their plausible decision-making process and what responsibilities they had addressed. The recorded interview conversations were reviewed by the medical authors of this article to itemize their clinical responsibilities and decision-making process. We pilot-tested this interview protocol with two residents who did not participate in the first study.

For the first interview study, we conducted in-depth individual interviews lasting around two hours, recognizing the sensitive nature of the research questions. The choice to start with interviews was guided by our aim to gather rich, contextual data about neurology residents' personal experiences and decision-making processes. These narratives are vital to appreciating the complex dynamics of competence and confidence within a clinical setting. Eleven neurology residents from three resident training hospitals (HU, KKU, KU) in Seoul were recruited, utilizing a combined purposeful and convenient selection strategy. We prioritized interviewing senior trainee doctors (third and fourth-year residents) because their experiences span both the old and new curricula, providing valuable insights into the impact of the curricular shift on competence and decision-making. The interviews were carried out by RYU (a non-medical author of this article), who did not know all the interview participants before the interview and had no hierarchical relationship with the residents to avoid any professional conflict. Potential participants were personally invited through messages or in-person requests to participate voluntarily in the interview. RYU.Y, another non-medical author of this article, briefly introduced the study, explaining the data to be collected and the interview process. All eleven invited individuals willingly agreed to participate and provided informed written consent. However, three

out of the eleven participants had to drop out due to unforeseen emergency calls during the interview. The interviews were video-recorded and transcribed verbatim (RYU. Y carried out this with the support of a Speech-to-Text application) with identifying details anonymised. All interviews were conducted in Korean. The interview occurred from November 13th to November 27th, 2022. The data were subsequently reviewed and analyzed for research purposes between November 28th, 2022, and December 18th, 2022.

The second study employed a cross-sectional design to compare the neurology residents' self-assessment with the evaluations their supervisors provided regarding the clinical tasks they performed. The study included a total of 44 neurology residents and eight supervisors from five resident education hospitals in Korea. Within the purview of our study, around 14 per cent (44 out of a total of 320 neurology residents in Korea) would be representative to contrast neurology residents' self-assessments against evaluations provided by their supervisors. To assess their competencies, the residents were requested to complete the K-NEPA13 (Korea Neurologist's Entrustable Professional Activities 13), a self-assessment tool that measures their proficiency in thirteen clinical tasks (please refer to S1 Appendix). It is important to note that one of the competencies, specifically Question 10 in S1 Appendix, applied only to senior residents. The supervisors were also tasked with completing the corresponding K-NEPA13 for each resident they supervised. In cases where multiple supervisors were involved, their evaluations were averaged to facilitate further analysis. RYU.Y (non-medical author) complied with the data to be blinded for further analysis. RYU (non-medical author) had the subsequent analysis on the compiled data. All individuals (residents and supervisors) who participated in the assessment process agreed to participate and provided informed written consent before data collection. The self-assessment data from the residents were voluntarily collected during the first two weeks of February 2023, immediately following the Neurology specialty examination held on February 1st, 2023. The evaluations of the residents by their supervisors were also conducted within the initial two weeks of February 2023. The two datasets were reviewed in the third week of February 2023 to ensure their completeness and confirmed for further analysis.

Finally, the third study intended to corroborate and extend the findings from the first two studies, using a survey administered across multiple resident training sites in Korea. Here, the sample of 97 neurology residents (from a pool of 320 registered in the National Neurology Residents division) was chosen to capture a wider range of resident experiences and enhance our findings' generalizability. The mixture of junior and senior residents (1st– 17, 2nd– 23, 3rd– 27, and 4th– 30) further allowed for an exploration of differing perspectives based on their respective stages in the residency program. Before beginning the survey, all participants agreed to provide their responses, a commitment formalized by a written consent form included at the start of the online survey. They were also informed of their freedom to withdraw from the study at any point if they chose to. The survey was conducted from March 1st to March 14th, 2023, spanning two weeks. The data were accessed on March 16th for research purposes.

Although the three studies were performed in sequence, the presentation of the studies does not follow a specific order for an easier readership. That is, we are not intentionally separating each analysis (interview, self-assessment, and survey), otherwise arranging them to effectively represent the findings.

## Data analysis

The interview data underwent an iterative and collaborative analysis process. Initially, RYU reviewed all anonymized transcripts and developed preliminary codes. Through regular meetings, four researchers (RYU, OH, KIM and CHOI) engaged in discussions, examining the

transcripts and the initial code list. The analysis with the code primarily focused on the residents' clinical competencies and responsibilities, exploring how their competence influenced their decision-making in their clinical responsibilities.

The assessment data on K-NEPA13 (see S1 Appendix) from both supervisors (8) and residents (44) in the five resident training university hospitals (HU, KKU, KU, IU, and CNNU) were directly compared on the simple linear regression (ideally, the supervisor evaluation and the resident evaluation are the same or perfectly aligned). Additionally, inter-rater reliability was assessed when supervisors provided different evaluations for the same residents. The mean scores of the supervisors' evaluations were then employed for further analysis.

The survey questionnaire (see S2 Appendix) used in the third study was primarily derived from the codes developed in the first study, with additional considerations based on the insights gained from the second study regarding the challenges and issues of the new competency-based training. A total of 97 anonymized responses were gathered from various university training sites to ensure diversity. To validate and strengthen the understanding from the previous two studies, suitable statistical tests were employed depending on the nature of the questions.

Note that the funder (Korea Health Industry Development Institute, The Ministry of Health & Welfare, Republic of Korea) had no role in study design, data collection and analysis, the decision to publish, or the preparation of the manuscript.

## Results

Our analyses revealed three key findings. Firstly, junior residents who have been exclusively educated through the new CBME curriculum recognized the significance of having their clinical decisions validated by supervisors. However, this counterintuitively resulted in a perceived overestimation of their professional competencies, which would be accompanied by overconfidence. It is important to acknowledge that these findings are based on cross-sectional data, and no causal relationship can be established to directly attribute these interpretations to the new competency curriculum. Nonetheless, it is evident that the new curriculum in Korea still needs to fully meet the objectives set by the global competency movement.

Conversely, senior residents demonstrated a relatively weak confidence in their clinical tasks. This can be attributed to the complex and heavier nature of their clinical responsibilities and the increased autonomy in decision-making without constant supervisor involvement. While this lack of enhanced competence is understandable, there is a concern if it underestimates their capabilities. Interestingly, the senior residents also tended to maintain consistency in their decision-making process, as evidenced by their reluctance to actively seek out information that challenges their existing beliefs. This inclination towards consistency hampers their motivation for critical rethinking.

Residents' core competencies for future career development vary based on their specific career paths. Our last finding indicated a shift in the emphasis on core competencies, such as clinical decision-making and ethical concerns, as residents progress through their years of residency. In the first two years, residents prioritize acquiring clinical knowledge, while in the latter two years, there is a noticeable shift towards placing greater importance on ethical concerns and collaboration with peers, ranking them higher. This evolving perception of competencies highlights the need for a comprehensive and adaptable competency program to meet changing demands and practice settings. The lack of comprehensive competence training poses challenges for residents in developing the necessary skills to become independent specialists in the future. Therefore, addressing overconfidence, consistency bias, and aligning with career development demands should be focal points in the future competency-based education program.

## Overconfidence in junior residents: The 'Dunning-Kruger' effect

The Dunning-Kruger effect is a cognitive bias that can have significant implications for early career development [19]. This bias occurs when individuals with limited ability or knowledge overestimate their competence. This overestimation can lead to errors and mistakes, particularly in tasks or responsibilities that require a certain level of expertise. For instance, a new employee may mistakenly believe they can complete a task without realizing their limitations, which can result in problems or subpar performance. Additionally, individuals influenced by the Dunning-Kruger effect may struggle to recognize their own weaknesses and may be reluctant to seek assistance when necessary. While it is important not to automatically assume the presence of the Dunning-Kruger effect in all training contexts, exploring its potential influence within the newly implemented competency-based curriculum for the neurology resident training in Korea is relevant. This examination will shed light on whether the new competency curriculum inadvertently promotes or mitigates the impact of this cognitive bias.

To examine any Dunning-Kruger effect among our residents, a total of 44 participants (20 junior residents and 24 senior residents) from five university hospitals (HU, KKU, KU, IU, and CNNU) completed self-assessments using the K-NEPA 13 (Korean Neurologist's Entrustable Professional Activities 13, see S1 Appendix) tool. At the same time, their eight supervisors were also asked to provide their own evaluations. For some residents, more than one supervisor assessed, the interrater reliability Kappa value was then calculated (0.83), indicating a relatively high level of consistency in the evaluations provided by the supervisors.

Fig 1 illustrates that junior residents tend to exhibit a stronger inclination towards the better-than-average effect in all competencies, although to varying degrees (particularly overestimating understanding care system and independence). On the other hand, senior residents demonstrated an underestimation effect, particularly in communication with patients and collaboration with peers. It is important to note that first and second-year residents received

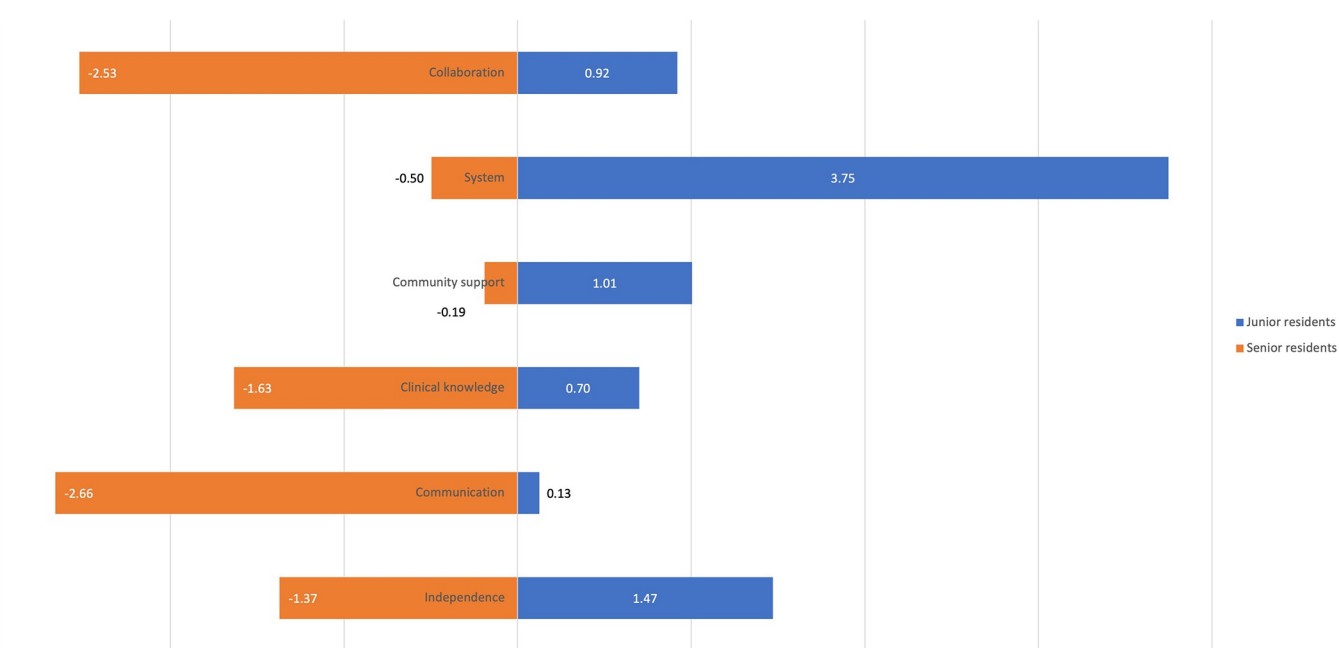

**Fig 1. The difference between self-evaluation vs supervisor evaluation on the six competencies (Collaboration with peers, understanding care system, community support, clinical knowledge, communication skills, and independence, refer to S1 Appendix): + means overestimation,—for underestimation.**

exclusive training through the CBME curriculum, while senior residents experienced a mix of the old and new curricula. The junior residents' exclusive training with designated supervisors may have influenced their perception of clinical work-based training, although caution is necessary as it may lead to an inflated perception of independence. In contrast, senior residents, engaged in more independent decision-making and complex clinical responsibilities (including supervising junior residents), exhibited an underestimation effect, potentially due to the lack of full supervisor support. These differing self-evaluations of competencies highlight the urgent need for a more dynamic supervisor intervention strategy, taking into account the specific competencies that are overestimated or underestimated.

We detailed these findings in Fig 2. Generally, as residents gain more experience, their competencies tend to improve. This is evident in the face values of the six competencies, which are consistently higher for senior residents compared to junior residents. However, it is important to highlight that junior residents tend to exhibit a higher degree of overconfidence when compared to the evaluations provided by supervisors. These findings offer valuable insights into the new competency-based program and its potential for future reforms. While the exclusive competency training showed promise in shaping positive confidence among junior residents, it is essential to maintain a balance and ensure that they do not overestimate their abilities (or underestimate for senior residents). This is particularly crucial in clinical knowledge, which significantly impacts patient safety. Additionally, the findings suggest the need for the reformed curriculum to align more closely with competencies such as understanding the care system and fostering independence, where the sample distributions exhibit wider variation. By addressing these competency areas, the competency program can be refined to better cater to the training needs of residents and enhance their overall competence in a broader range of areas.

## The interview data indicates a tendency to confuse confidence with competence

The findings discussed above were further corroborated by the interview data. The senior residents generally presented their clinical competence as common and potentially appreciated, with one resident stating,

> "*I think in those patient cares and realistic training experiences are vital to our education. So, I think I come to know more on what I have to learn and what I don't know, otherwise, it would make everything harder, and we would not be able to improve*" (Senior Resident 4).

Senior residents expressed a common theme of fearing mistakes, as evidenced by the quotes from Senior Residents 1 and 2. They acknowledged the importance of clinical experiences but felt significant pressure to avoid errors. This fear may impact their perception of competencies, particularly independence, as they may be inclined to choose safer and more conventional decisions. However, as previously mentioned, providing adequate supervision and feedback can help alleviate this concern and enhance residents' confidence and competence.

> "*I think clinical workplace experience will take us much further and bring a lot of learning benefits. But when I think I had mistakes I am terrified, so not to do it again is my first priority (. . .) In recent, I had to make a decision about either psychogenic nonepileptic seizures or symptomatic ones on the EEG reading. . . it was hard to clear-cut on the reading, so I did take a more conservative approach*" (Senior Resident 2)

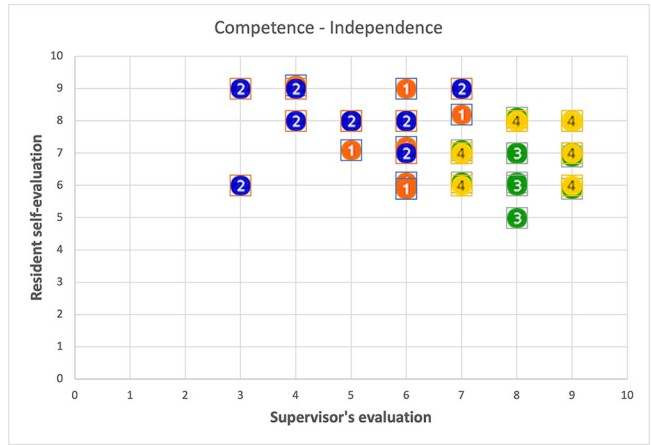

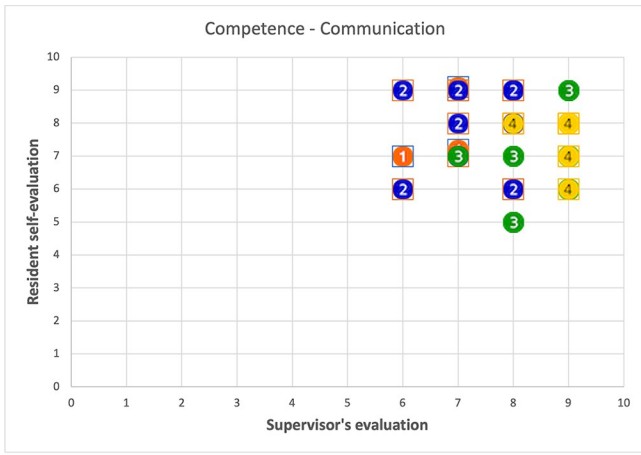

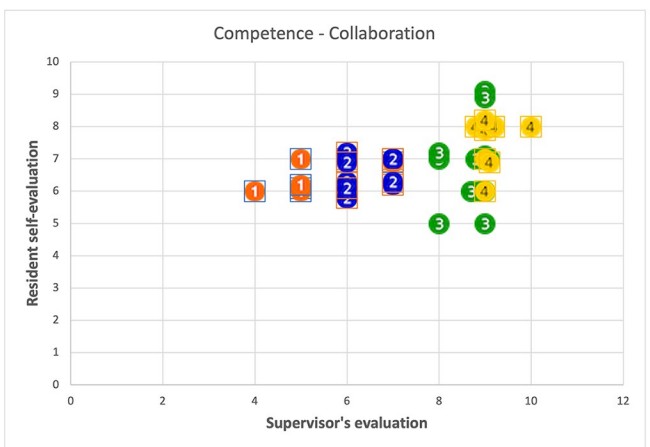

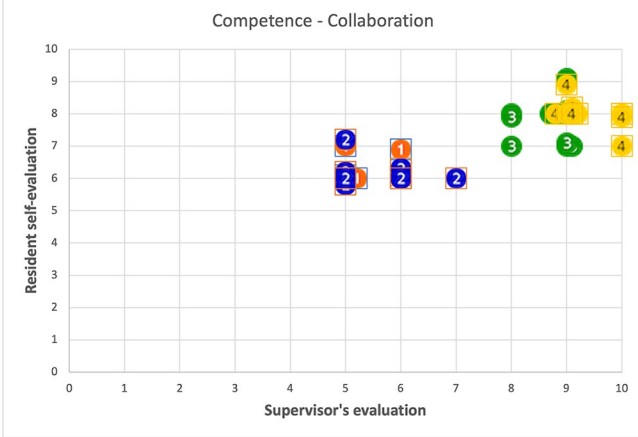

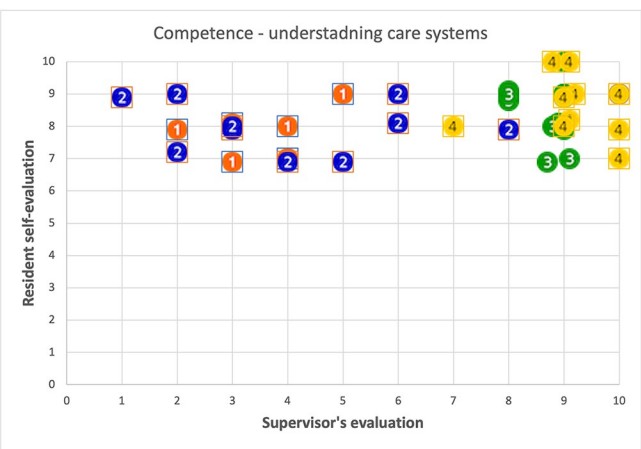

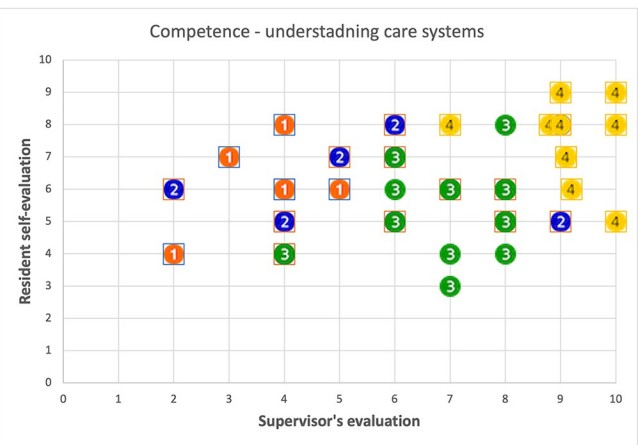

**Fig 2. Competencies (cyan–junior 1st year; blue–junior 2nd year; green–senior 3rd year; yellow– 4th year): The data point above the dotted line means overconfidence.**

"*I think it's good. I have learnt a lot of things in the hospital, and my supervisor allows me to do this by myself, but it is always stressful not to make any mistakes. It is my privilege, but whether I can do this on my own is still not sure. (. . .) sometimes, it is much easier because I*

*have seen this before, and such patient cases seem much more common, so I can easily do something on these patients (. . .) (hmm) if (there are) some rare cases, I need to look at some internet before asking it to the supervisors. I am somewhat worried if they judge me on this.*" (Senior Resident 1)

In contrast, the junior resident seems more conviction in their competence-based learning, and more frequent interventions from the supervisors is thus necessary.

"*I always feel the weight of responsibility on my shoulders when making clinical decisions, especially when it's a complex case. I acknowledge that I have gained a significant amount of knowledge and skills during my training from the supervisor, but I still worry about missing something important or overlooking crucial factors. It's a constant battle between feeling confident in my abilities and being aware of my limitations. My initial decisions would be often more incorrect. There are instances where I need to engage in critical rethinking. I must reassess and adapt my initial plans. But it is too overwhelming and asks more of the supervisor.*" (Junior Resident 4)

These came with difficult emotional responses. Senior residents shared that they felt "bad, guilty, angry and . . . shame" (Senior Resident 1), "disappointed" (Senior Resident 4), "anxious" (Senior Resident 2), and "horrible" (Senior Resident 1), when it came to know they were wrong. This can be attributed in part to their increased autonomy in decision-making, which makes them more susceptible to experiencing remorseful feelings when their decisions result in negative patient outcomes. Consequently, this leads to diminished confidence in their actual competencies. Many senior residents shared stories of self-blame and shame regarding their past mistakes. As expressed by Senior Resident 2, "The way to overcome this unpleasant feeling is to be perfect and just work tirelessly" (Senior Resident 2). Consequently, this mindset contributes to their underestimation of their competence. Senior Resident 3 vividly illustrates this perspective.

"*I know that carefully reviewing patients' medical histories and tests, and need to second opinions from supervisors. I also recognize the significance of continuous self-reflection to identify areas where I can enhance my knowledge, skills, and approaches to patient care(. . .). However, I am the most senior resident here, so I am also responsible for supervising junior residents, which necessitates me being more knowledgeable of what I am thinking.*" (Senior Resident 3)

## The survey data shows the need for reform of the current competency program

Based on the results obtained from both selective interviews and the cross-sectional study on K-NEPA13 for self-assessments, it was evident that the implementation of the competency curriculum in our residency program has had an impact, highlighting an urgent need for specific adjustments. To gain further insights into residents' perceptions of the new competence curriculum, a survey was conducted among 97 neurology residents, representing a response rate of 36% out of the total registered 320 neurology residents in Korea.

Participants were prompted to rank (from the most important to least) the core competencies they considered crucial for attaining independence as neurology specialists. These core competencies were aligned with the reflections of the Korean Neurological Association's discussions in 2022, and the medical authors of this article actively participated in ensuring the

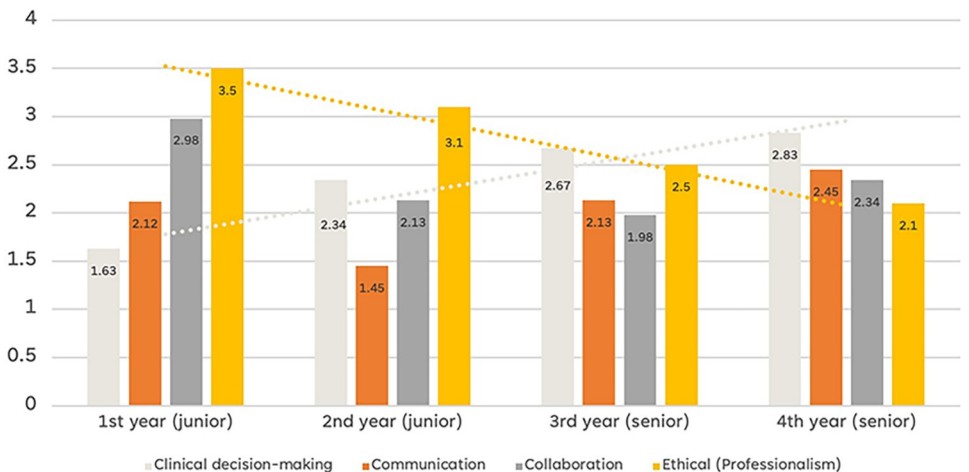

**Fig 3. Ranked core competence: Mean ranks (s.d).**

inclusion of these competencies for neurological specialists. Notably, ethical concerns, as a component of professionalism, reflect the commitment to promoting the health and well-being of individuals and society through ethical practices, profession-led regulations, and high standards of personal behavior and clinical practice. These specific competencies identified from the codes in the interview data are considered for reforming the current competence curriculum, serving as the foundational framework for the future residency training program.

The data presented in Fig 3 highlight a shift in the emphasis on the two core competencies (i.e., clinical decision-making and ethical concerns) among the residents based on their year of residency. It indicates that in the initial two years, residents prioritize acquiring clinical knowledge more. However, in the latter two years, there is a notable shift towards placing greater importance (i.e., lower rank) on ethical concerns and collaboration with peers. This observation aligns with our previous discussions regarding the residents' evolving perception of competencies throughout their training, highlighting the need for a comprehensive and adaptable competency program. Of particular interest is the growing emphasis on ethical concerns for final-year residents. This is also supported by an interview conducted with a fourth-year resident, who shared their perspective on the importance of addressing chronic conditions in her future practice:

> "*As I prepare to establish my own clinic in the community, I anticipate encountering a different patient population compared to the more critical and urgent cases I have encountered at the university hospital. Therefore, I believe it is crucial to enhance my residency training in areas that pertain to addressing the needs of community-based patients and managing chronic conditions.*" (Senior Resident 2)

This interview highlights the need to adapt the competency program to meet the changing demands and practice settings that residents will encounter upon graduation. This lack of comprehensive competence training poses potential challenges for residents in developing the necessary skills for their future career development. Second-year residents also require proficiency in communication with patients as their primary responsibilities involve caring for in-hospital patients. On the other hand, the third-year residents need to focus on collaboration with their peers, as their clinical responsibilities ask for more competencies that can serve the emergency care setting. These specific demands underscore the significance of providing

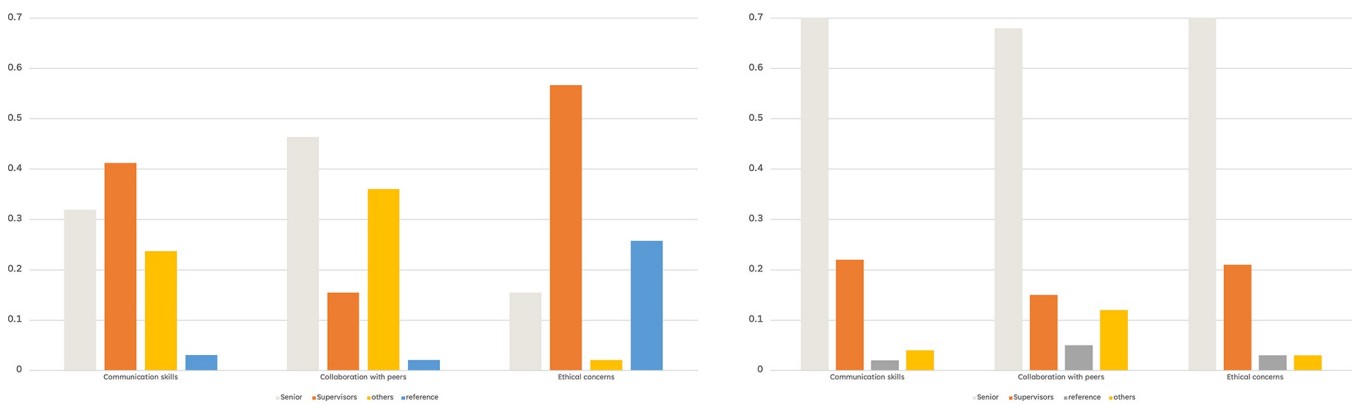

**Fig 4. Core competence learning: What shapes the emphasis?** Junior residents (left) Senior residents (right).

residents with the essential skills and knowledge to deliver comprehensive care in diverse healthcare settings.

Fig 4 presents the contrasting learning experiences of junior and senior residents in acquiring core competencies as assessed by the survey items. The results indicate that junior residents, who exclusively underwent the new residency program and had designated supervisors, primarily relied on their supervisors' guidance to acquire core competencies not covered in the old curriculum. In contrast, senior residents predominantly learned these competencies from their fellow senior residents rather than relying on their supervisors. These findings partially confirm the effects of the new residency program, particularly in introducing designated supervisors to facilitate the acquisition of relevant competencies.

Four survey items (Table 1) were used to evaluate residents' self-perceived readiness in handling complex clinical scenarios that require the application of core competencies. Considering the previous discussion on K-NEPA13, the responses to these survey questions also suggested that the core competencies of junior residents were generally comparable to those of senior residents, which implies some level of overconfidence, indicating an urgent need for further alignment in the future residency program. The results of independent sampled t-tests did not reveal any significant differences between the two groups.

## Consistent bias: Insights for intervening in resident learning process

Building upon the findings of our previous cross-sectional study, we have uncovered that competence and confidence do not exhibit a straightforward linear relationship. This discovery raises concerns regarding the potential impact of overconfidence within the new residency competence program and underscores the need for reform. In our quest to delve deeper into this issue, we turn our attention to a cognitive bias known as consistency bias. This bias manifests as favoring information that aligns with existing beliefs or attitudes. It is a common

**Table 1. The perceived level of preparedness mean (s.d).**

|  | Junior residents | Senior residents | sig |
|---|---|---|---|
| **Clinical decision-making** | 3.81 (1.28) | 4.52 (0.98) | n.s |
| **Communication with patients/ caregivers** | 3.72 (1.32) | 4.64 (1.23) |  |
| **Collaboration with peers** | 3.99 (0.64) | 4.28 (1.21) |  |
| **Ethical concerns** | 3.83 (1.23) | 4.63 (0.99) |  |

phenomenon observed among experts who frequently engage in similar tasks, as their perception of consistency can bolster their expertise [20]. That said, clinical expertise development has heavily relied on linear cause-and-effect thinking, which assumes a direct and consistent relationship between a cause and its effect. However, this linear approach may prove inadequate or even misleading in many medical conditions often involving multiple interacting factors. Consequently, effective medical reasoning critical in the residency education program necessitates rethinking and flexible reasoning against a simple cause-and-effect linear thinking [21]. The following interview data vividly illustrate this:

"*I've learned to think more critically about cause and effect when making clinical decisions. I take into account all the possible factors that could be contributing to a patient's condition and consider the potential consequences of my decisions. . . But also I am to be humble enough to seek input from my supervisors and senior residents and consult the latest research*" (Junior Resident 2)

"*(. . .) There were many different factors to consider, and it wasn't always clear which one was the most important. For example, I recently had an emergent patient who presented with symptoms that could be attributed to either a stroke or a seizure. I wasn't sure which one it was, so I consulted with my supervisor, who helped me think through the different possibilities and come up with a plan.*" (Junior Resident 1)

"*I was assigned to a patient who presented with sudden-onset severe headache and confusion. I suspected a subarachnoid hemorrhage and started the appropriate treatment. However, as the patient's condition worsened despite treatment, I began to re-evaluate my initial diagnosis and management plan. At the same time, I consulted with my supervisor and we discussed other possible causes, such as cerebral venous thrombosis or encephalitis. We together adjusted the treatment plan accordingly and the patient eventually stabilized.*" (Junior Resident 4)

In contrast, senior residents may establish more confident cause-and-effect relationships due to their significantly higher experiences:

"*I tend to think in a more simplistic cause-and-effect manner when it comes to patient care. I focus on identifying the real problem and then finding a quick and most effective solution, rather than considering all the potential factors and complexities that could be involved.*" (Senior Resident 4)

"*When I encounter a patient with symptoms that fit a specific neurological pattern, I can quickly diagnose and treat them. For example, I recently saw a patient with sudden onset of severe headache, nausea, and vomiting, which made me think of subarachnoid hemorrhage. I ordered a CT scan and lumbar puncture, which confirmed my suspicion. I was able to provide prompt treatment and transfer the patient to the neurosurgery department. Similarly, when I see a patient with sudden onset of weakness on one side of the body, I immediately suspect a stroke and order a brain MRI. My strong cause-and-effect reasoning enables me to make quick and accurate diagnoses, which is essential in the fast-paced environment of a hospital.*" (Senior Resident 2)

## Supervisors motivate rethinking: The interview data

Overcoming this consistency bias commonly observed in the experts is difficult; however, we can employ some teaching strategies to mitigate its impact. One effective approach is to cultivate awareness of the bias and actively remain open to new information, even if it contradicts

preexisting beliefs [22]. This openness allows for a more balanced and objective evaluation of evidence. The focus of this section is that if residents can be aware of their consistency bias towards linear cause-and-effect reasoning, or the supervisors can detect this bias; they can take steps to address it with appropriate educational strategies.

Our residents recognized feedback from supervisors on their clinical decision-making as a crucial supervisory strategy. One senior resident suggested: "*I think it happens all the time. I think, actually, it's vital* (Senior Resident 1)," whereas another resident explained that "*I had time pressure to call appropriate treatment decisions that I was not allowed to get feedback from supervisors*" (Senior Resident 4).

However, the residents who were allowed to rethink struggled to cope with the iteratively long thinking process; for instance, they needed to figure out when to stop rethinking.

"*When it comes to treating patients, there is no simpler process. Take reducing pain, for example. It looks simple, but in reality, there are numerous factors to consider, and it's not always clear when to stop rethinking or overthinking. I want to make sure we're making the right decision, but there's always that feeling of doubt*". (Junior Resident 2)

"*I don't really have that problem anymore. I've seen so many cases over the years that I usually know what to do right away. I don't need to keep rethinking things.*" (Senior Resident 4)

Although they offered many personal examples of needy rethinking, senior residents, in particular, did not explicitly reveal that they communicated openly about it with supervisors. Interestingly, the junior residents who have been exposed to the new residency program (i.e., designated supervisor) had more easily talked to their supervisors.

"*I've been practicing for years, and I've seen it all. I trust my gut instinct and my experience. I don't need to second guess myself every time I make a decision. Take stroke patients for example. I've seen so many cases that I know exactly what to do, and I don't need to waste time rethinking my approach. Of course, I often consult with my colleagues for more complex cases, but for the most part, I'm confident in my ability to make the right call.*" (Senior Resident 2)

"*if (there are) some rare cases, I need to look at some internet before asking it to the supervisors. I am somewhat worried if they judge me on this.*" (Senior Resident 1)

"*I am lucky to have a fully dedicated supervisor who strongly encourages me to adopt a mindset of trying to find other alternatives. She told me that neurological disorders are complex and multi-faceted, requiring a more flexible and nuanced approach. For instance, when treating a patient with a seizure disorder, it's important to consider not only the patient's medical history and current symptoms, but also any potential triggers, such as stress or sleep deprivation, that may be contributing to their seizures.*" (Junior Resident 1)

Our data found numerous instances where junior residents reported receiving guidance from their supervisors emphasizing the importance of rethinking or updating their beliefs, even in time-pressured situations. Without such communication, trainees were left to independently interpret the symptoms. Notably, senior residents appeared to receive less frequent communication of this nature within their current educational strategy.

### Resident education needs a dynamic approach: The survey data

To validate the findings from the interview data and gain a broader perspective on a reformed future residency program, a survey was conducted among 97 neurology residents in Korea.

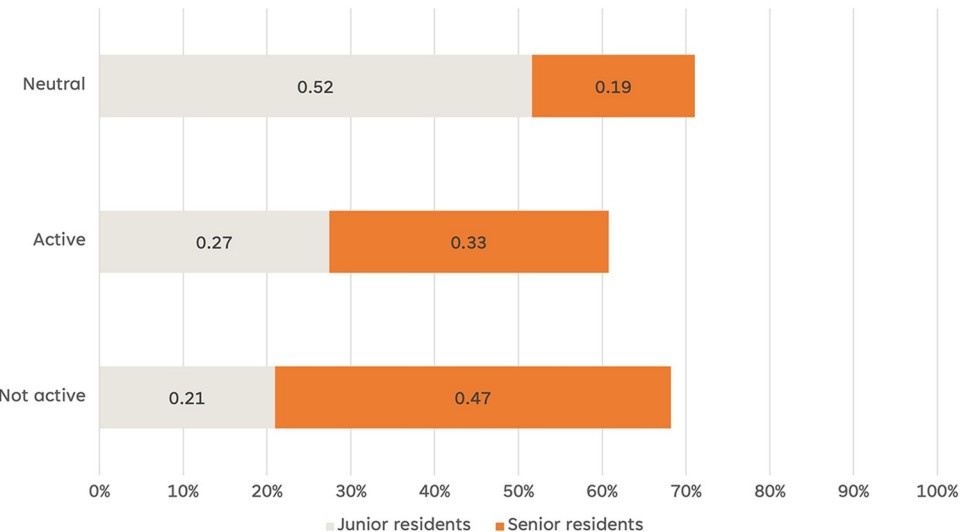

**Fig 5. The survey question on consistency bias: The senior residents trust more their initial decisions.**

The survey aimed to explore the residents' clinical decision-making process and whether it challenges their initial beliefs. The survey included a specific question asking residents about their inclination towards aligning with their initial decision or seeking alternative perspectives when making clinical decisions. Fig 5 presents the distribution of responses, revealing that 47% of senior residents preferred aligning with their initial decision, while only 21% shared the same sentiment.

Although this data represents the residents' retrospective views on their decision-making process, it provides valuable insights. The substantial proportion of senior residents who did not seek second opinions suggests that consistency in their thought process is more welcomed and reinforced than being challenged. It can partly arise from their extensive clinical experiences and partly from their heavy clinical responsibilities. These interpretations highlight the importance of implementing a more balanced and dynamic education strategy that considers reducing the clinical responsibilities of senior residents. It is crucial to note that the survey captures residents' self-reported tendencies and does not directly assess their actual decision-making behavior. These survey findings, otherwise, underscore the need for a dynamic teaching strategy that encourages residents to explore alternative perspectives, challenge their initial beliefs, and cultivate a more flexible and adaptive approach to clinical decision-making.

While the response to the survey question alone provides valuable insights, it is important to acknowledge that it does not definitively establish the presence of consistency bias among senior residents. To gain a deeper understanding, an additional survey question was included to explore the reasons behind their inclination to either reconsider or maintain their initial decision-making approach. Fig 6 illustrates the responses to this question, shedding light on the residents' perspectives. A substantial number of senior residents expressed a preference for maintaining their usual approach, indicating a tendency towards adhering to established patterns of decision-making. Notably, approximately half of the senior residents reported that they continued to make decisions in the same manner as they have always done. In contrast, a significant portion of junior trainees (around a quarter) voiced concerns about overthinking, suggesting a potential hesitancy to critically reassess their initial decisions. These differing attitudes towards rethinking and overthinking highlight the need for a nuanced and dynamic educational strategy that addresses the challenges senior and junior residents face.

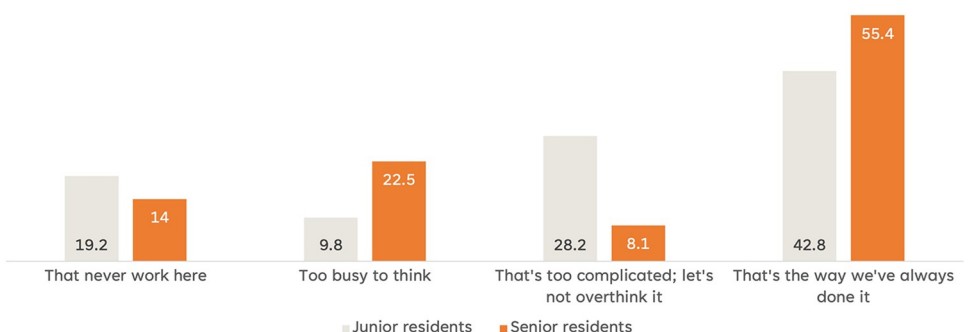

**Fig 6. Junior residents worry the overthinking (28.2%), and seniors present a status-quo thinking (55.4%).**

Our findings underscored the need for dynamic educational strategies to encourage critical thinking and foster a culture of openness to alternative viewpoints in clinical decision-making processes.

## Discussion

In general, confidence goes hand in hand with competence. However, our important finding was that the junior residents seemed to overestimate their actual competence, especially arising from the frequent supervisor's intervention as part of the merit of the new residency program. Our findings run counter to previous research [23, 24], showing that confidence gets intensified as they have more clinically experienced, and current supervising interventions for the residents fail to balance their actual competence and confidence. This research provides valuable insights for the ongoing development and improvement of the residency training program, emphasizing the importance of maintaining a balanced perspective on residents' abilities and aligning the educational strategy with key competencies, particularly for some institutions that aim to adopt the global competency movement.

Though the overconfidence problem of the junior residents can be carefully managed via appropriate supervisor's intervention, the primary concern would direct to the senior residents who do not have significant interventions from the supervisors. We found that updating their previous knowledge discussed in Section 3.2 is pivotal. The core nature of one's cognitive flexibility is non-linear thinking, and accordingly, supervisory interactions with these resident groups is essential in the resident training program [21]. Patel et al. [25] argued that challenging common beliefs and avoiding a simple and linear "cause and effect" are the main educational practices in the clinical context.

These findings highlight the significance of developing a more comprehensive educational strategy that considers the varying demands of residents at different stages of their training. In addition to the dynamic teaching strategy, it is essential to incorporate career development education into the new residency program. By recognizing and actively addressing these challenges for competency-based resident education, medical educators and practitioners can foster a culture of reflection and adaptability, ultimately enhancing the effectiveness of training programs and improving overall patient outcomes.

## Limitations

Our study is subject to several limitations that should be considered. Firstly, the interview sample size (8) was small; therefore, the findings may not represent all neurology residents in Korea. This selective interview was inevitable not to disrupt the participating residents' clinical activities by finding available time slots. However, it is important to note that this limitation

was partially addressed by including additional data from diverse hospitals through the K-NEPA assessment. This approach may mitigate the potential impact of the small interview sample size and provide a broader perspective on the residents' competencies. Secondly, our study design was primarily cross-sectional, preventing the establishment of causal relationships. Thus, it is not possible to determine whether the discrepancies observed between junior and senior residents are directly attributed to the new residency program. The absence of a controlled experimental approach and the lack of a comprehensive assessment of the entire residency program using the same research methodology employed in this study might hinder causal inference. Thirdly, the study did not evaluate the residents' actual clinical competencies, such as using 360 evaluation, as this would require a quantitative and experimental study design, contrasting with the qualitative nature of our investigation.

Our study design presents limitations that should also be acknowledged, impacting the generalizability and depth of our findings. The utilization of selective interviews as our data collection method provides valuable insights into the perceptions and interpretations of the residents. However, it also restricts the comparability of experiences among interviewees and may result in retrospective and linear expressions of their clinical experiences. Future research incorporating alternative data collection approaches could enhance the comprehensiveness of our insights. Regarding the assessment of K-NEPA13, the evaluation process relied on the residents' and supervisors' holistic and reflective judgment rather than direct observational or inquiry-based competence assessment. Improvements in the assessment methodology, such as incorporating more objective observational methods, could further refine our understanding of trainees' competence in relation to K-NEPA13. Additionally, our sampling strategy did not consider particular training contexts, such as regular supervisor-trainee meetings, to discuss learning progress against predefined outcomes. Future studies could adopt a more robust sampling approach to capture a variety of training contexts, enabling a more comprehensive exploration of the influence of specific contextual features.

These limitations provide opportunities for future research to overcome methodological constraints and delve deeper into the intricacies of residents' experiences and competence development.

## Conclusions

Residents recognize that their experience is not perfect. Therefore, medical supervisors need to foster a culture of continuous learning and self-reflection by encouraging them to actively seek feedback and embrace opportunities for personal growth. Adopting a proactive and personalized supervisory approach can mitigate the negative effects of overconfidence and consistency bias in the current residency training curriculum and facilitate safer patient care.

In the pursuit of cultivating essential competencies and cognitive skills, it is evident that the current competency-based training curriculum may need to be revised. Our study emphasizes the significance of incorporating an awareness of cognitive biases when designing medical education strategies. Medical educators and practitioners can foster a culture of continuous learning and self-reflection among trainees by actively addressing these biases and fostering an adaptable approach within competency-based training programs. Moreover, it is important to consider the influence of cultural factors on consistency bias, especially within the context of heavy clinical workloads in residency programs. Further research exploring these cultural factors and their variations across different medical education settings holds substantial value in enhancing our understanding of this phenomenon.

In conclusion, our research not only broadens the understanding of competency-based training but also enriches the theoretical framework of CBME. Our findings will thus stimulate

critical discussions and contribute to designing more effective supervisory strategies within the global medical education community, thereby enhancing resident training and patient outcomes in diverse contexts.

## Supporting information

**S1 Appendix. K-NEPA 13 stands for Korea Neurologist's Entrustable Professional Activities 13.** It is an assessment tool that measures residents' competence in their ability to perform 13 critical clinical tasks (competencies-Communication, Collaboration, Clinical knowledge, independence, Community support and understanding care *System*).
(DOCX)

**S2 Appendix. The survey items (8) used in this study.**
(DOCX)

## Acknowledgments

We thank our participants for trusting us to present their time and efforts in the interviews and surveys.

## Author Contributions

**Conceptualization:** Hojin Choi, Jeeyoung Oh, Hokyoung Ryu.

**Data curation:** Hojin Choi, Hokyoung Ryu, Youngji Ryu.

**Formal analysis:** Hojin Choi, Hokyoung Ryu, Youngji Ryu.

**Funding acquisition:** Hojin Choi, Hokyoung Ryu.

**Investigation:** Jeeyoung Oh, Chi Kyung Kim, Hokyoung Ryu.

**Methodology:** Hokyoung Ryu.

**Project administration:** Hokyoung Ryu.

**Resources:** Jeeyoung Oh, Chi Kyung Kim, Hokyoung Ryu.

**Software:** Hokyoung Ryu, Youngji Ryu.

**Validation:** Hokyoung Ryu.

**Visualization:** Hokyoung Ryu, Youngji Ryu.

**Writing – original draft:** Hojin Choi, Jeeyoung Oh, Hokyoung Ryu.

**Writing – review & editing:** Chi Kyung Kim, Hokyoung Ryu.

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
