## [Decision Letter · Decision Letter 0]

20 Jul 2023

PONE-D-23-17722Residents need competence not confidence: A retrospective evaluation of the new competency education program for Korean neurology residentsPLOS ONE

Dear Dr. Ryu,

Thank you for submitting your manuscript to PLOS ONE. After careful consideration, we feel that it has merit but does not fully meet PLOS ONE’s publication criteria as it currently stands. Therefore, we invite you to submit a revised version of the manuscript that addresses the points raised during the review process.

Please adjust you paper according to Reviewers remarks.

We look forward to receiving your revised manuscript.

Kind regards,

Radoslaw Wolniak, full professor

Academic Editor

PLOS ONE

Journal Requirements:

"CH/RH - Grant number : HC22C0014

Korea Health Industry Development Institute (KHIDI)

The Ministry of Health & Welfare, Republic of Korea

No"

5. Please upload a new copy of Figure 2 as the detail is not clear. Please follow the link for more information: " ext-link-type="uri" xlink:type="simple">https://blogs.plos.org/plos/2019/06/looking-good-tips-for-creating-your-plos-figures-graphics/"
" ext-link-type="uri" xlink:type="simple">https://blogs.plos.org/plos/2019/06/looking-good-tips-for-creating-your-plos-figures-graphics/"

Reviewers' comments:

Reviewer's Responses to Questions

**Comments to the Author**

1. Is the manuscript technically sound, and do the data support the conclusions?

Reviewer #1: Yes

Reviewer #2: Yes

2. Has the statistical analysis been performed appropriately and rigorously? 

Reviewer #1: Yes

Reviewer #2: Yes

3. Have the authors made all data underlying the findings in their manuscript fully available?

Reviewer #1: Yes

Reviewer #2: Yes

4. Is the manuscript presented in an intelligible fashion and written in standard English?

Reviewer #1: Yes

Reviewer #2: Yes

5. Review Comments to the Author

Reviewer #1: The paper addressed an interesting topic. The theoretical section is quite acceptable but needs to discuss recent literature. Below are some recommendations to enhance the quality of this research:

The abstract is poorly structured and far too long. Abstract: should be written again in academic forms. Main aim, where study conducted, model used, period, and main findings.

Do not cite detailed results of the study. Just outline - it is an abstract.In the abstract, I propose to emphasise the aim of the work, the research question posed or the hypothesis. The abstract must focus on objectives, mention how they were achieved, and emphasize the results obtained. The abstract is confusing.

Introduction: The author (s) should highlight the contribution of this work to the existing literature review in the last part of the introduction. The written contribution is not clear

Research Methods: The reasonableness of the sample selection should be declared.

Reviewer #2: Article at a very good level of content. The structure of the study is correct. Very interestingly designed empirical study and professionally conducted analysis of the results. The only suggestion for a possible addition concerns the expansion of the theoretical part concerns the description of the analyzed areas of competence in the empirical research part.

6. PLOS authors have the option to publish the peer review history of their article (what does this mean?). If published, this will include your full peer review and any attached files.

Reviewer #1: No

Reviewer #2: No

---

## [Author Response · Author response to Decision Letter 0]

30 Jul 2023

In the cover letter, I have fully addressed the comments raised by both reviewers and editor.

---

## [Editor Report · Decision Letter 1]

10 Aug 2023

Residents need competence not confidence: A retrospective evaluation of the new competency education program for Korean neurology residents

PONE-D-23-17722R1

Dear Dr. Ryu,

We’re pleased to inform you that your manuscript has been judged scientifically suitable for publication and will be formally accepted for publication once it meets all outstanding technical requirements.

Kind regards,

Radoslaw Wolniak, full professor

Academic Editor

PLOS ONE
---

## [Editor Report · Acceptance letter]

26 Sep 2023

PONE-D-23-17722R1 

Residents need competence not confidence: A retrospective evaluation of the new competency education program for Korean neurology residents 

Dear Dr. Ryu:

I'm pleased to inform you that your manuscript has been deemed suitable for publication in PLOS ONE. Congratulations! Your manuscript is now with our production department. 

Kind regards, 

on behalf of

Professor Radoslaw Wolniak 

Academic Editor

PLOS ONE